# Retention of Antioxidants from Dried Carrot Pomace in Wheat Bread

**Rafał Ziobro [1],\*** , **Eva Ivanišová [2]** , **Tatiana Bojňanská [2]** and **Dorota Gumul [1]**

1   Department of Carbohydrate Technology and Cereal Processing, University of Agriculture in Krakow, 30-149 Krakow, Poland
2   Faculty of Biotechnology and Food Sciences, Slovak University of Agriculture in Nitra, 94976 Nitra, Slovakia
\*   Correspondence: rafal.ziobro@urk.edu.pl

**Featured Application: The retention of antioxidant activity from plant material in wheat bread proved to be high. Therefore, the authors suggest using carrot-enriched breadcrumbs as a carrier of health-promoting substances for culinary use.**

**Abstract:** The trial aimed to check the retention of carrot pomace antioxidants in wheat bread, using a variety of colorimetric assays. It was observed that the addition of 15% dried carrot pomace significantly affected dough properties. The enriched bread was acceptable in terms of technological parameters but exhibited strong carrot flavor and orange color. The incorporation of antioxidants from carrot pomace allowed us to increase the antioxidant potential of wheat bread (32% for DPPH and ABTS assays, 3 times for FRAP and 4 times for FOMO). The extent of the changes in contents of individual groups of antioxidants was not always proportional to the applied addition level (increase in polyphenols was almost eight times, flavonoids—nine times, phenolic acids—two times and flavonols—eight times, compared to the control). The study revealed that the antioxidant properties of the pomace are largely preserved during baking, and therefore such a product could be used for the production of breadcrumbs for coating and admixing purposes.

**Keywords:** carrot pomace; antioxidants; wheat bread

## 1. Introduction

Post-production waste from vegetable juice processing can be a source of valuable antioxidant compounds [1], which have repeatedly proven hypoglycemic, hypocholesterolemic, anticancerogenic anti-inflammatory, antiviral, antimicrobial, anti-allergic, anticoagulant effects, reduce post-prandial glycemia, hypertension, and the risk of diseases such as atherosclerosis and other cardiovascular diseases, cataracts, diabetes, genetic damage, degenerative bone changes, and neurodegenerative diseases including Alzheimer's disease [2–4]. One of the most valuable sources of such compounds may be the pomace produced during the manufacture of carrot juice.

Carrots are an important source of so-called phytonutrients, i.e., compounds of plant origin (mainly secondary metabolites) with health-promoting properties. This group includes a number of phenolic compounds and carotenoids. The phenolic acids found in carrots are mainly derivatives of hydroxycinnamic acid [5].

In terms of vitamins, thiamin, riboflavin, niacin, folic acid and vitamin C are present in significant amounts. The anthocyanin content varies according to color, reaching, in black varieties, 1750 mg/kg [6,7].

The high levels of carotenoids significantly shape the therapeutic properties of carrots, which include a diuretic effect, and the maintenance of proper nitrogen levels by removing excess uric acid. Carrot consumption can also have a positive effect on the immune system, protecting against heart attacks, hypertension, osteoporosis, cataracts, arthritis, asthma and urinary tract infections [6,7]. Nevertheless, the storage of carrots for longer periods results

in significant changes in color, total soluble solids, acidity, total phenolics, carotenoids and antioxidant activity [8], which is one of the reasons for its industrial processing.

Phenolic compounds in carrot roots are unevenly distributed, with the highest amount found in the outer parts. The total content of phenolic compounds depends on the variety, and varies in studies by different authors, depending on the methodology used. According to Alasalvar et al. [5] the total amount of phenolic acids (after summing up the individual acids determined by chromatography) in purple carrots was 74.64 mg/100 g, while in other varieties it ranged from 7.72 to 16.21 mg/100 g. Similarly, Roszkowska et al. [9] observed the highest content of phenolic compounds (1930 mg per 100 g d.m. in terms of catechin) in purple carrot root and the lowest (806 mg per 100 g d.m.) in white carrot root. Even higher values for orange carrots (1400–1582 mg in 100 g d.m.) were recorded by Cieślik et al. [10].

According to Nawirska and Kwaśniewska [11], the fiber substances of carrots include pectin (7.41% on a dry weight basis), hemicelluloses (9.14%), cellulose (80.94%) and lignin (2.48%). Carrots are therefore a rich source of calcium pectinate, which has been associated with the ability to lower cholesterol levels and liver enzyme activity [12] consequently reducing the risk of hypertension, heart attack, heart disease and cancer [13].

The water content of carrots ranges from 86–89% [6], resulting in yields of 60–70% during juice production. Only a fraction of the health-promoting compounds end up in the juice, as, for example, the amount of carotene remaining in the pomace can be as high as 80% [14]. The antioxidant activity and content of bio-active compounds in carrot waste may be of importance in health promotion, prevention and therapy [15]. It has been shown that their levels in carrot juice change due to various processing methods, such as blanching, enzyme liquefaction and pasteurization [16], so similar effects could be expected for the pomace. Fresh carrot pomace is unstable in terms of microbial spoilage and should be quickly dehydrated by a range of drying methods [17].

Dried carrot pomace could be used in a variety of food products of plant and animal origin including bread, cake, dressings, pickle, cookies, and functional drinks [7], but also various extrudates [18] and a range of meat products [19]. The use of carrot pomace for enrobing deep-fried products was suggested, and compared with traditional breading systems [20], giving some promising results. The application of battered and breaded coatings for supplying antioxidants in meat products, especially those designed for deep-frying, has not been thoroughly studied yet, although lipid oxidation is one of the key reactions affecting their quality. An attempt to incorporate quercetin at a concentration of 1% into the edible coating in order to increase the oxidative stability of chicken drumsticks proved unsuccessful [21], probably due to its low concentration. It seems that no trials were made to enrich the breading systems by supplying the bread with antioxidants.

Wheat flour is one of the staple foods that can easily be combined with various types of industrial food by-products for bread manufacture in quantities up to 20% [22], although smaller amount are usually applied to maintain sensory acceptance [23]. There are no data on the antioxidant content and activity in cases of a large addition of carrot pomace to bread. Thus, the aim of the study was to assess the influence of 15% addition of carrot pomace on antioxidant content and potential of wheat bread.

## 2. Materials and Methods

### 2.1. Material

The study material consisted of commercially available carrot (Amplus sp. z o.o., Poland), wheat flour 650 (PZZ Kraków, Poland), freeze-dried yeasts (Saf-instant, S.I. Lesaffre, France), and salt.

### 2.2. Methods

Fresh carrot pomace after juice extraction (juicer Zelmer 177.T, with a yield around 60%) was placed on the sheets of filter-paper (approx. 0.5 mm height) and dried in a sterilizer (Memmert SF-55, Memmert GmbH + Co. KG, Schwabach, Germany) at 60 °C for



24 h. The dried pomace was finely ground using a laboratory mill (Perten LM 3100, Perten Instruments, Hagersten, Sweden) and stored in glass jars.

Farinographic evaluation of the flour type 650 and its mix with 15% dried carrot pomace was made using Brabender farinograph-E (Brabender Technologie GmbH & Co. KG, Duisburg, Germany), according to the AACC method 54–21 [24].

The bread was prepared from the dough consisting of wheat flour mix (in the case of the CP sample, 15% of flour was replaced with dried carrot pomace), yeast (2% on flour basis) and water (depending on water absorption). Dough pieces (approx. 300 g) was proofed for 40 min at 35 °C and baked for 40 min at 200 °C (MIWE Condo; MIWE Michael Wenz GmbH, Arnstein, Germany). After baking, the loaves were removed from baking pans, cooled to ambient temperature, cut into slices (5 mm. thick), dried overnight under ambient conditions, ground into crumbs and stored in glass jars for further analyzes.

- Chemical evaluation

Determination of dry matter, ash and protein was carried out in accordance with the AACC method 08-01 [24].

The nitrogen content was determined by the semi-micro-Kjeldahl method. A conventional factor of 5.7 for bread and 6.25 for carrot pomace was used to convert nitrogen into protein.

Fat content was assessed with the Ancom XT15 Fat Extractor (USA). To remove moisture before extraction, the sample (1.5 g, W1) was weighed into a special filter bag (XT4, Ankom Technology, Macedon, NY, USA)) and dried in an oven (WTB, Binder GmbH, Tuttlingen, Germany) at 105 °C for 180 min, after which it was cooled in a desiccant pouch for 15 min and weighed again (W2). Petroleum ether extraction was carried out at 90 °C for 60 min. The defatted samples were dried in an oven at 105 °C for 30 min, cooled in a desiccant pouch and weighed again (W3). The following formula was used to calculate the fat content (%): $[(W2 - W3)/W1] \times 100$.

Ethanol extracts used for the determination of antioxidant constituents were obtained by dissolving 0.6 g of the sample in 30 mL of ethanol (80 g/100 g). The mixture was shaken for 120 min (electric shaker: type WB22, Memmert, Schwabach, Germany), and centrifuged (15 min, 4500 rpm. $1050 \times g$) in a centrifuge type MPW-350, (MPW MED. Instruments, Warsaw, Poland). The supernatant was decanted and stored at −20 °C for further analyses.

- Total content of polyphenols was determined with Folin–Ciocalteau reagent, using a spectrophotometer, according to Singleton et al. [25]. 5 mL of the extract was diluted to a volume of 50 mL with distilled water. 5 mL of the diluted extract was combined with 0.25 mL of Folin–Ciocalteau reagent (previously diluted with distilled water in the proportion 1:1 $v/v$), 0.5 mL 7% $Na_2CO_3$. The contents were vortexed (WF2, Janke & Kunkel, Staufen, Germany) and stored for 30 min in a dark place. The absorbance was measured using Helios Gamma 100–240 (Thermo Fisher Scientific, Runcorn, UK), at the wavelength $\lambda = 760$ nm. The results were converted to mg catechin/100 g d.m.
- Determination of flavonoids was performed by the spectrophotometric method, according to El Hariri et al. [26], 0.5 mL of the extract was combined with 1.8 mL distilled water and 0.2 mL 2-aminoethyldiphenylborate reagent in a test tube. The contents were vortexed, and the absorbance was measured at the wavelength $\lambda = 404$ nm. Flavonoid content was expressed as mg of rutin/100 g d.m.
- Contents of total polyphenols (TPC-NFC, without F-C reagent), phenolic acids, flavonols and anthocyanins were determined by spectrophotometric method, according to Mazza et al. [27], with the modification of Oomah et al. [28]. 0.1 mL of the extract was mixed with 2.4 mL of 2% HCl in 75% in a test tube. The contents were vortexed and the absorbance was measured in at the wavelength $\lambda = 280$ nm (TPC), $\lambda = 320$ nm (phenolic acids), $\lambda = 360$ nm (flavonols) and $\lambda = 520$ nm (anthocyanins). TPC was expressed in mg of catechin/100 g d.m., phenolic acids in mg of ferulic acid/100 g d.m., flavonols in mg quercetin/100 g d.m., and anthocyanins in mg glycoside-3-cyanidin/ 100 g d.m.

- Free-radical scavenging activity

Free-radical scavenging activity of samples was measured using 2,2-diphenyl-1-picrylhydrazyl (DPPH) by method of Sánchéz-Moreno et al. [29]. The ethanol extract of carrot pomace (0.4 mL) was mixed with 3.6 mL of DPPH solution (0.025 g DPPH in 100 mL ethanol). In the case of bread samples, the extracts (1 mL) were mixed with 4 mL of DPPH solution (0.025 g DPPH in 100 mL ethanol).

Absorbance of the reaction mixture was determined at 515 nm using the spectrophotometer Jenway (6405 UV/Vis, Jenway Ltd., Dunmow, UK). Trolox (6-hydroxy-2,5,7,8-tetramethylchroman-2-carboxylic acid) (10–100 mg/L; $R^2 = 0.989$) was used as the standard, and the results were expressed in mg/g Trolox equivalents.

- Antiradical activity

The radical scavenging activity was measured by the ABTS [2,2′-azino-bis (3-ethylbenzothiazoline-6-sulfonic acid)] radical cation discoloration assay, as described by Re et al. [30]. ABTS was dissolved in water to a 7 mM concentration. ABTS radical cation (ABTS$^{+\bullet}$) was produced by reacting ABTS stock solution with 2.45 mM potassium persulfate (final concentration) and allowing the mixture to stand in the dark at room temperature (12–16 h) before use. The bleaching rate of ABTS$^{+\bullet}$ in the presence of the sample was monitored at 734 nm using a Helios Gamma 100–240 (Thermo Fisher Scientific, Runcorn, UK) spectrophotometer. The ABTS$^{+\bullet}$ solution was diluted in PBS buffer (pH 7.4) to give absorption value of $0.700 \pm 0.05$ for the analysis of extracts. Volumes of 2.00 mL of ABTS$^{+\bullet}$ and respective extracts in PBS buffer solution were used. The ABTS$^{+\bullet}$ bleaching was monitored at 30 °C, and the discoloration after 6 min was used as the measure of antiradical activity. Radical scavenging activity was measured as Trolox equivalent antioxidant capacity (mg of Trolox per g of sample). Trolox solutions used for calibration curve were used in the concentration range 10–100 mg/L ($R^2 = 0.9957$).

- Ferric reducing antioxidant power (FRAP)

The reducing power of samples was determined by the method of Oyaizu [31]. One milliliter of sample extract was mixed with 5 mL PBS (phosphate buffer with pH 6.6) and 5 mL of 1% potassium ferricyanide in the test tube. The mixture was stirred thoroughly and heated in water bath for 20 min at 50 °C. After cooling, 5 mL of 10% trichloroacetic acid was added. A volume of 5 mL of mixture was pipetted into the test tube and mixed with 5 mL of distilled water and 1 mL of 0.1% ferric chloride solution. Absorbance at 700 nm was measured using the spectrophotometer Jenway (6405 UV/VIS, Jenway Ltd., Dunmow, UK). Trolox (10–100 mg/L; $R^2 = 0.9974$) was used as the standard, and the results were expressed in mg/g Trolox equivalents.

- Molybdenum reducing antioxidant power (FOMO)

The molybdenum reducing antioxidant power of samples was determined by the method of Prieto et al. [32], with slight modifications. The mixture of sample (1 mL), monopotassium phosphate (2.8 mL, 0.1 M), sulfuric acid (6 mL, 1 M), ammonium heptamolybdate (0.4 mL, 0.1 M) and distilled water (0.8 mL) was incubated at 90 °C for 120 min, then rapidly cooled and detected by monitoring absorbance at 700 nm using the spectrophotometer Jenway (6405 UV/VIS, Jenway Ltd., Dunmow, UK). Trolox (10–1000 mg/L; $R^2 = 0.998$) was used as the standard, and the results were expressed in mg/g Trolox equivalents.

- Total carotenoid content

An amount of 1 g of sample was homogenized in the mortar and repeatedly extracted with 10 mL acetone until the sample became colorless. The extract was filtered using a Whatman No. 1 filter paper and used for detection of total carotenoid content.

Petroleum ether placed in a separation funnel with a Teflon plug, acetone sample extract and distilled water were poured along the walls. The mixture was separated into two phases, and the aqueous phase was discarded. The residual acetone was removed by

washing the petroleum ether phase twice with distilled water. To remove the residual water, the petroleum ether phase was passed through a small funnel containing 5 g of anhydrous sodium sulfate into a 50 mL volumetric flask.

The contents of the flask were topped up with petroleum ether, and the total carotenoid content was determined from the molar absorption coefficient of β-carotene [33]. The concentration (mg/g) of carotenoids was calculated using the formula:

$$\text{TCC}\left[\text{mg} \cdot g^{-1}\right] = \frac{A \cdot r \cdot V \cdot 10}{E \cdot n}$$

where: *A*—absorbance at 445 nm; *r*—sample dilution; *E*—molar absorption coefficient $E^{1\%}_{1cm} = 2620$; *n*—sample weight; TCC—total carotenoid content.

- Statistical analysis

The experimental data were subjected to analysis of variance (Duncan's test), at the confidence level of 0.05, by the use of software Statistica v. 8.0 (Statsoft, Inc., Tulsa, OK, USA). All measurements were done in duplicate, except for ash content ($n = 4$) and dry matter ($n = 3$).

## 3. Results and Discussion

### 3.1. Dough Properties

Water absorption wheat flour type 650 mixture with 15% dried carrot pomace was significantly higher than in its absence (Table 1). The addition of pomace resulted, on the one hand, in a decrease in the water content of the prepared mixtures from 12.7 to 11.7% and, on the other hand, in an increase in the content of fiber substances, effectively binding water. This outcome was reflected in a significant (15%) increase in water absorption, which was also accompanied by a significant elongation of dough development time and increased stability. The trends were similar to those observed earlier for wheat flours supplemented with carrot pomace [34,35].

**Table 1.** Farinographic properties of wheat flour (WF) and its mixture with 15% carrot pomace (CP).

| Property | Wheat Flour | WF + 15% CP |
|---|---|---|
| Moisture content [%] | 12.7 ± 0.1 [b] | 11.7 ± 0.1 [a] |
| Water absorption [%] | 54.0 ± 0.1 [a] | 63.7 ± 0.3 [b] |
| Development time [min] | 1.5 ± 0.1 [a] | 6.0 ± 0.3 [b] |
| Stability [min] | 4.3 ± 0.1 [a] | 7.3 ± 0.1 [b] |

Different letters in the row mean statistically different average values ($\alpha = 0.05$).

### 3.2. Bread Properties

Baking of the dough containing carrot pomace, in which more water was added, resulted in a more pronounced baking loss (Table 2). The resulting loaves baked were slightly smaller than the control but retained acceptable specific volume. Nevertheless, the organoleptic properties of the bread were unacceptable in terms of intensive orange color and carrot smell. This is not strange when taking into account earlier reports in which the optimum addition level of carrot pomace in terms of sensory assessment was 5% [23], 2.5% [36] or up to 3% [35].

### 3.3. The Influence of Carrot Pomace on Bread Properties

#### 3.3.1. Chemical Composition of Bread

The first stage of the study concerned determinations on carrot pomace (Table 3). Under the applied drying conditions, moisture content was around 6%, while carbohydrates predominated among the components of the dry matter. In the case of bread, the moisture content was approximately 13% for both samples. The content of protein in carrot pomace was above 8%, a little higher than the 6.7% reported by Kohajdowa et al. [35], but significantly less than in the bread, which seemed little affected by the applied formulation.

**Table 2.** Basic characteristics of control wheat bread and wheat bread enriched with 15% carrot pomace (CP).

| Property | Wheat Bread | CP Bread |
|---|---|---|
| Weight | 281.75 ± 2.22 [b] | 271.75 ± 4.64 [a] |
| Volume | 556.99 ± 4.20 [b] | 435.39 ± 1.77 [a] |
| Specific volume | 1.98 ± 0.03 [b] | 1.60 ± 0.03 [a] |
| Baking loss | 12.65 ± 0.07 [a] | 16.05 ± 0.35 [b] |

Different letters in the row represent statistically different average values ($\alpha$ = 0.05).

**Table 3.** Basic characteristics of the carrot pomace and bread.

| Property | Carrot Pomace | Wheat Bread | CP Bread |
|---|---|---|---|
| Dry matter [%] | 94.1 ± 0.12 [c] | 79.4 ± 0.17 [b] | 74.4 ± 0.16 [a] |
| Protein [% d.m.] | 8.32 ± 0.99 [b] | 13.15 ± 0.17 [a] | 13.09 ± 0.60 [a] |
| Fat [% d.m.] | 1.386 ± 0.179 [c] | 0.155 ± 0.064 [a] | 0.259 ± 0.127 [b] |
| Ash [% d.m.] | 4.16 ± 0.05 [c] | 0.87 ± 0.11 [a] | 1.21 ± 0.10 [b] |

Different letters in the row mean statistically different average values ($\alpha$ = 0.05).

On the other hand, the fat content was significantly higher in carrot pomace, as compared to the bread, although it was a little lower than the 2.1% reported by the above-mentioned authors [35].

The content of ash was almost three times higher than in the aforementioned study [35], but less than 5.5% higher than what was reported by Tańska et al. [23]. A significant amount of minerals found in the carrot pomace affected their levels in bread. The ash contents reported for fresh carrot varieties may change significantly. In the study [37] on twelve carrot varieties, the values of ash content changed from 5.18 to 9.91%, which could be regarded as the upper limit for its content in dried pomace, because the soluble minerals are likely to be removed with the juice. The partial replacement of wheat flour (containing 0.65 % ash) with dried carrot pomace caused an adequate increase in ash contents of the bread.

3.3.2. Antioxidants in Bread

The presence of main groups of antioxidants was determined by colorimetric assays (Table 4). The content of total polyphenols depended on the applied assay method, including the type of extraction, and could be expressed in various ways (e.g., various phenolic acids used to calculate polyphenols [38–40].

**Table 4.** Major groups of antioxidants in carrot pomace, control wheat bread and wheat bread enriched with 15% carrot pomace (CP).

| | Carrot Pomace | Control Bread | CP Bread |
|---|---|---|---|
| TPC-FC [mg catechine/g] | 428.88 ± 20.74 [c] | 22.56 ± 0.11 [a] | 170.17 ± 11.12 [b] |
| Flavonoids [mg rutin/g] | 72.26 ± 17.00 [c] | 3.02 ± 0.20 [a] | 28.32 ± 2.07 [b] |
| TPC-NFC [mg catechine/g] | 372.36 ± 36.11 [c] | 16.49 ± 0.01 [a] | 151.21 ± 16.69 [b] |
| Phenolic acids [mg ferulic acid/g] | 20.77 ± 2.08 [c] | 3.09 ± 0.01 [a] | 5.48 ± 0.01 [b] |
| Flavonols [mg quercitin/g] | 14.34 ± 1.14 [b] | - | 8.34 ± 1.27 [a] |
| Carotenoids [mg/g] | 0.56 ± 0.02 [b] | - | 0.048 ± 0.002 [a] |
| Anthocyanins | - | - | - |

Different letters in the row mean statistically different average values ($\alpha$ = 0.05).

The content of polyphenols in carrot pomace is substantial—428.88 mg catechin/100 g, and according to Roszkowska et al. [9] (1930 mg per 100 g d.m. in terms of catechin) in purple carrot root and the lowest (806 mg per 100 g d.m.) in white carrot root. Even higher values for orange carrots (1400–1582 mg in 100 g d.m were determined by Cieslik et.al. [10]. According to Tarko et al. [41], although phenolic compounds present in carrot are represented mainly by chlorogenic, p-hydroxybenzoic, caffeic and cinnamic acid derivatives, the total content of polyphenols in carrot pomace is low (12 mg catechin/100 g fresh mass) in comparison to other types of pomaces (grape, apple, red beet). The main reason for this is probably generally low content of dry mass constituents and high moisture content. Also, according to Sánchez-Rangel et. al. [42] carrot pomace obtained immediately after juice extraction contained 1530 mg chlorogenic acid equivalents per kilogram. Thus, carrot pomace contains relatively small amounts of total phenols compared to other plant materials such as blueberries, purple potatoes and cherries, where their content is 41,800, 7810 and 20,980 mg/kg, respectively [43].

Analyzing the contents of polyphenols in carrot pomace, it could be stated that, according to many authors [38–40], there are many factors which could influence the final results of polyphenol determination in the sample. They include the applied method of analysis, extraction conditions, including the extractor composition, temperature and time of extraction, proportion between solids and liquids and storage conditions. Also, the variety, plot location and agronomic conditions could influence the total content of polyphenols in plants [44,45]. The juicing method and processing conditions are the final factors for establishing pro-health properties of the pomace. It has been shown that wounding stresses and UV radiation could increase the level of polyphenols both in fresh-cut carrot [46] and in processed pomace [42].

The flavonoid content of carrot pomace is at 72.26 mg rutin/100 g d.m., while phenolic acids account for only 20.77 mg per ferulic acid. According to Sánchez-Rangel et al. [42], chlorogenic acid is the only phenolic compound identified in carrot pomace immediately after juice extraction, indicating that other phenolic compounds normally present in carrot roots (protocatechuic acid, derivatives of gallic acid, 3,5-dicaffeoylquinic acid, 3-hydroxy-dihydro chlorogenic acid, 4,5-dicaffeoylquinic acid, p-coumaric acid and ferulic acid) are removed with the juice [47,48]

The content of polyphenols in control bread was on the level of 22.56 mg catechin/100 g, which seems typical for cereal products.

Nevertheless, the control bread proved to be deficient in polyphenols, while the addition of carrot pomace during its preparation caused a significant increase in their level, much larger (almost 8 times) than could be expected from the level of addition. This may suggest that the baking step is partly responsible for the increase in polyphenols. Indeed, it is known that Maillard reactions occurring under high temperature result in the increase of TPC values.

The content of total polyphenols in the control bread is most likely to be explained not only by the polyphenols from the wheat flour but also by the Maillard reaction products present in it, which could influence the results because of the side reactions with a broad range of constituents [49,50]. According to Katina et al. [51], fermentation could also increase the proportion of polyphenols in the final bread. The confirmation of this influence was earlier reported by Korus et al. [52] found, who measured the content of total polyphenols in starch-based gluten-free bread at the level of 5.2 mg catechin/100 g d.m. Similar increase in the level of polyphenols were earlier reported for gluten-free bread with an addition of apple pomace [53].

The volume of flavonoids, phenolic acids and flavonols in breads with 15% addition of carrot pomace was 9 times, 2 times and 8 times higher compared to the control, respectively. An unexpected increase (higher than 15% pomace addition) of the above-mentioned polyphenol compounds can be explained by the concerted action of many factors. Most likely, this is related to the subsequent stages of bread production, because, according to Katina et al. [51] the quantity of phenolic acids increases during the yeast fermentation

process, as well as during dough mixing. In addition, this may in part be due to the thermal breakdown of quercetin derivatives, especially quercetin rutinoside, which generates phenolic acids [54–56]. In the case of flavonols, which include quercetin derivatives according to Rupasinghe et al. [55], their heat resistance is relatively high compared to phenolic acids and anthocyanins. Nevertheless, the baking temperature used (230 °C) may have led to their partial breakdown, as evidenced by the appearance of the aforementioned phenolic acids in the breads.

In conclusion, it should be stated that the baking process affects the loss of some phenolic compounds [57,58]. According to the authors, losses of these compounds can reach up to 60%. These losses are influenced by a great number of factors, such as thermal, enzymatic and oxidative degradation, as well as the isomerization/epimerization and decarboxylation processes of phenolic acids mentioned earlier [58,59].

Despite the fact that thermal processes such as baking contribute to the loss of polyphenols, the carrot pomace introduced in finely ground form proved to be a valuable source of bioactive compounds, providing a significant content of phenolic compounds in the final product. Another reason for the elevated health-promoting properties of bread with carrot pomace could be the formation of new antioxidant compounds, i.e., products of the Maillard reaction and transformation of polyphenolic compounds at various stages of bread making.

On the other hand, the increase in carotenoids was not as big as the addition of carrot pomace; it could be observed that only half of carotenoids added in the formulation was extracted from the bread. The loss could be ascribed mainly to the baking step. In earlier studies on carrot pomace, the amounts of total carotenoids varied largely from 5 to more than 100 mg/100 g d.m. [15]. Some of this variation is probably due to different juicing and drying methods, which are known to affect the level of carotenoids. The obtained value (56 mg/100 g d.m.) is in the middle of this range.

Thus, the use of carrot pomace in breads has significantly increased the proportion of health-promoting substances (polyphenols and carotenoids) which exhibit, among other things, anti-inflammatory, anti-allergic, anticoagulant, antiviral and anti-cancer effects [60–62]. There is a lack of use of carrot pomace in baked goods in the literature, the exception being the following publication.

Powdered carrot pomace dried in the microwave or on heat was included in the cookies (replacing 30% of wheat flour) to increase their phytochemical content by Hernandez-Ortega et al. [63]. Cookies with the addition of microwave-dried carrot pomace revealed the highest levels of β-carotene, epicatechin, gallic acid and ferulic acid compared to the control.

### 3.3.3. Antiradical Properties of Bread

The significant content of antioxidants from polyphenolic groups together with carotenoids was reflected in the results of the antioxidant potential of bread (Table 5). Antioxidant activity is determined by the chemical structure of these compounds, in particular, the type, number and location of functional groups in the molecule, the concentration of the compounds and their interactions with other food components (which is particularly true for polyphenols) [64,65].

**Table 5.** Free-radical scavenging activity (DPPH, ABTS), Ferric reducing antioxidant power (FRAP), Molybdenum reducing antioxidant power (FOMO), Antioxidant activity of carrot pomace, control wheat bread and wheat bread enriched with 15% carrot pomace (CP).

|  | Carrot Pomace | Control Bread | CP Bread |
|---|---|---|---|
| DPPH [TEAC] | 4.98 ± 0.25 [c] | 2.94 ± 0.08 [a] | 3.89 ± 0.32 [b] |
| ABTS [TEAC] | 19.57 ± 0.21 [c] | 11.35 ± 0.13 [a] | 14.96 ± 0.11 [b] |
| FRAP [TEAC] | 39.67 ± 0.66 [c] | 1.94 ± 0.18 [a] | 7.56 ± 0.12 [b] |
| FOMO [TEAC] | 243.39 ± 2.42 [c] | 20.02 ± 1.42 [a] | 64.63 ± 1.62 [b] |

Different letters in the row mean statistically different average values ($\alpha = 0.05$).

According to Tarko et al. [41] α- and β-carotene are the most valuable compounds of carrot pomace, being the precursor of vitamin A in human organism. Apart from this physiological role, carotenoids are known for their ability to react with singlet oxygen and free radicals, which allows us to include them among natural antioxidants [66]. Their high level (0.56 mg/g d.m.), allows us to predict that they could significantly affect antioxidant properties of the pomace. The significant content of antioxidant groups was reflected in the results of DPPH scavenging activity and antioxidant power (Table 5). All these methods are generally based on the single electron transfer mechanism, but DPPH radical may also be neutralized by the hydrogen atom abstraction [67]. The smallest differences between carrot pomace and wheat bread were found for free-radical scavenging activity, but also in the case of this parameter the addition of carrot pomace caused a significant increase in the values. FRAP values were 20 times lower for the control bread in comparison to carrot pomace, and the change in its formulation caused the value to increase almost four times.

FOMO values of carrot pomace were more than 10 times larger than for wheat bread and after its addition to bread recipe, a three-fold increase could be observed.

The observed increase in DPPH and ABTS radicals' scavenging activity, caused by the addition of carrot pomace to wheat bread, was about 32%, while the parameters FRAP (323% of the value for the control sample) and FOMO (389%) more than tripled. This may be to some extent connected with a different radical scavenging mechanism and uneven reducing activity of various compounds present in wheat bread. It should be added that the results of the DPPH analysis show similar trends, but not very clear relation to the results of the other analyses used to determine antioxidant potential (ABTS, FRAP and FOMO methods) This is most likely to be explained by the fact of interference of other compounds (for example, carotenoids), which are absorbed at the wavelength used in this determination [68]. The pomace, as mentioned earlier, is a good source of carotenoids, which are lipophilic and could be easily extracted with applied solvents. Their presence during the assay can generate errors in the determination of antiradical activity using the DPPH radical. Hence, the results of this determination are low, and the differences between the results for individual samples are small (compared to other methods).

It should be stressed that polyphenols act as antioxidants in several directions: as reducing agents, substances that deactivate free radicals, chelate metals catalyze oxidation processes, activate antioxidant enzymes and deactivate oxidative enzymes, etc. [69,70]. In food, all the above-mentioned ways of action of these compounds are encountered simultaneously. It can be concluded that the estimation of the antioxidant capacity of a given product is difficult, as it requires the use of multiple methods, based on different principles, taking into account the multiple roles of polyphenols as antioxidants. Also, the interpretation of the results of antioxidant activity determined by only one method is far from being sufficient and requires caution, especially in terms of health-promoting activity. It seems justified to analyze the antioxidant activity using several methods, which broadly define the antioxidant potential of raw materials, semi-products (carrot pomace) and products (CP bread). Thus, in order to evaluate the antioxidant potential of products in this work, we used methods that take advantage of the ability of antioxidants to deactivate stable free radicals (ABTS, DPPH) and to reduce metal ions to a lower oxidation state by the antioxidant under study (FRAP, FOMO) [71,72]. These methods are complimentary, because the FRAP method covers most antioxidant components in the sample, while the DPPH method detects only some of the most reactive ones, and the ABTS method gives intermediate values [71].

In addition, other substances, such as macromolecular Maillard reaction products formed at high temperatures during baking, which also exhibit antioxidant properties, may have contributed to the antioxidant capacity discussed above [60].

Summarizing, the bread obtained in the study was characterized by a high content of health-promoting compounds compared to the control; nevertheless, its sensory characteristics (intensive color and carrot smell) could not be accepted for the commercial product. Since the dough used for baking is characterized by acceptable technological properties

and the baking process itself runs smoothly, it will be possible to use the baking process to prepare a semi-finished product—breadcrumbs, suitable for coating and thickening applications, especially for enrobing of meat products. Such an ingredient, apart from its favorable nutritional composition, would be characterized by intense and desirable color and acceptable taste, with no change in processing technology in comparison to standard bread. The latter is especially important because of the significant influence of physical characteristics of breadcrumb coating on consumer acceptance [73]. Additionally, the use of such breading systems would broaden the range of natural antioxidants applicable for meat products [74].

## 4. Conclusions

Admixture of carrot pomace with the wheat flour resulted in its enrichment in ash and lipids. The combined flour changed its farinographic characteristics, significantly increasing its water absorption capacity. The use of carrot pomace in the production of wheat bread resulted in a significant loss of some carotenoids, but not the other groups of antioxidants. The overall antioxidant activity of the bread was significantly higher after such an addition, irrespective of the method of its determination. The influence of carrot addition was in some cases greater than the share of added carrot pomace, which indicates its synergistic effects with dough processing, and most probably the baking stage. The obtained breadcrumbs could be used in the manufacture of deep-fried battered and breaded products.

**Author Contributions:** Conceptualization, R.Z., E.I. and D.G.; methodology, E.I. and D.G.; software, R.Z.; validation, D.G., E.I. and T.B.; formal analysis, R.Z.; investigation, R.Z. and T.B.; resources, E.I. and D.G.; data curation, R.Z.; writing—original draft preparation, R.Z.; writing—review and editing, D.G.; visualization, R.Z.; supervision, D.G.; project administration, R.Z.; funding acquisition, R.Z. All authors have read and agreed to the published version of the manuscript.

**Funding:** This research received no external funding.

**Institutional Review Board Statement:** Not applicable.

**Informed Consent Statement:** Not applicable.

**Data Availability Statement:** Not applicable.

**Conflicts of Interest:** The authors declare no conflict of interest.

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
