# Peer review of "Retention of Antioxidants from Dried Carrot Pomace in Wheat Bread"

_applsci, doi:10.3390/app12199735_

Round 1

Reviewer 1 Report

Abstract: Include the numeric observation in this section.

Introduction: Include some information on shelf life of carrot thereafter explain what kind of methodology could be adopted to enhance the shelf life. Then explain your ideology.

Materials and method section: ABTS assay is discussed in brief. Explain the detailed methodology, including reagent preparation, incubation time. Use equation also. 

Results and discussion: Include the importance of different antioxidant assay used during the present study. 

Author Response

Abstract: Include the numeric observation in this section. 

The text of the abstract was supplemented with numerical data 

Introduction: Include some information on shelf life of carrot thereafter explain what kind of methodology could be adopted to enhance the shelf life. Then explain your ideology. 

Some information about the shelf life of carrots and carrot pomace was added. Nevertheless, the study was initially meant to prove that baking carrot-pomace enriched bread does not lead to excessive loss of antioxidant compounds (including polyphenols). Of course, we expect that the changes in antioxidant status during storage could be quite different for isolated pomace and breadcrumbs. In our opinion, the bread matrix will provide an alternative for stabilizing carrot antioxidants in solid food supports, which were earlier described for beta-carotene by Szterk and Lewicki (http://www.pttz.org/zyw/wyd/czas/2007,%205(54)/18_Szterk.pdf). The research presented in this paper focuses on how the addition of carrot pomace affects the polyphenol content of bread and what is the antioxidant potential of bread with a 15% CP addition. Research on changes in antioxidants and antioxidating potential of such bread, which we can recommend as breadcrumbs with increased health-promoting value, during its storage will be the subject of another publication.

Materials and method section: ABTS assay is discussed in brief. Explain the detailed methodology, including reagent preparation, incubation time. Use equation also.  

The methodology was corrected.  

Results and discussion: Include the importance of different antioxidant assay used during the present study.  

Appropriate changes were made in the text (section 3.3.3) 

Reviewer 2 Report

Corrections have been marked in the PDF for revision of MS. 

Author Response

line 38: citation is needed it is not your finding  

the citation was added 

line 49: It is in term of GAE or any other unit ??? 

the determination was done directly by HPLC for each detected compound 

line 121: rewrite this line for more clarity 

rewritten 

line 150: how you prepared methanolic extract what is concentration of CP in methanol 
and what is the concentration of bread in methanolic extraxt also??? mention  

Ethanol was used. The sentence was corrected.

line 180: write about filter number also 1, 2 or 42 etc. ??? 

Information was added .

line 210: also insert graph of farienographic analysis   

The graph (attached) will be added as a supplementary figure.

line 427: check as journal guide line  

The formatting has been revised 

Round 2

Reviewer 1 Report

The manuscript is significantly improved.  Congratulations!